# The Impact of Malnutrition on Chronic Obstructive Pulmonary Disease (COPD) Outcomes: The Predictive Value of the Mini Nutritional Assessment (MNA) versus Acute Exacerbations in Patients with Highly Complex COPD and Its Clinical and Prognostic Implications

**DOI:** 10.3390/nu16142303

**Published:** 2024-07-17

**Authors:** Domenico Di Raimondo, Edoardo Pirera, Chiara Pintus, Riccardo De Rosa, Martina Profita, Gaia Musiari, Gherardo Siscaro, Antonino Tuttolomondo

**Affiliations:** 1Division of Internal Medicine and Stroke Care, Department of Promoting Health, Maternal-Infant, Excellence and Internal and Specialized Medicine (ProMISE) “G. D’Alessandro”, University of Palermo, 90133 Palermo, Italy; edoardo.pirera@unipa.it (E.P.); chiarapintus1809@gmail.com (C.P.); derosariccardo96@gmail.com (R.D.R.); martinaprofita9@gmail.com (M.P.); gaiamusiari@gmail.com (G.M.); bruno.tuttolomondo@unipa.it (A.T.); 2PhD Programme “Molecular and Clinical Medicine”, Department of Promoting Health, Maternal-Infant, Excellence and Internal and Specialized Medicine (ProMISE) “G. D’Alessandro”, University of Palermo, Piazza Delle Cliniche 2, 90127 Palermo, Italy; 3Medical Affairs—Chiesi Italy SpA, 43122 Parma, Italy; g.siscaro@chiesi.com

**Keywords:** malnutrition, Mini Nutritional Assessment (MNA), chronic obstructive pulmonary disease (COPD), acute exacerbation of COPD (AECOPD), quality of life, frailty, multimorbidity

## Abstract

Background: Current management of COPD is predominantly focused on respiratory aspects. A multidimensional assessment including nutritional assessment, quality of life and disability provides a more reliable perspective of the true complexity of COPD patients. Methods: This was a prospective observational study of 120 elderly COPD patients at high risk of acute exacerbations. The Mini Nutritional Assessment (MNA) was administered in addition to the usual respiratory assessment. The primary outcome was a composite of moderate or severe acute exacerbations during 52 weeks of follow-up. Results: The median MNA Short Form (SF) score was 11 (8–12), 39 participants (32.50%) had a normal nutritional status, 57 (47.5%) were at risk of malnutrition and 24 (20%) were malnourished. Our multivariate linear regression models showed that the MNA score was associated with dyspnea and respiratory symptom severity, assessed by the Modified British Medical Research Council (mMRC) scale and the COPD Assessment Test (CAT) score, with spirometric variables, in particular with the severity of airflow limitation based on the value of FEV1, and with poorer QoL, as assessed by the EQ-5D-3 questionnaire. Competing risk analysis according to nutritional status based on the MNA Total Score showed that COPD participants “at risk of malnutrition” and “malnourished” had a higher risk of moderate to severe acute exacerbations with sub-hazard ratios of 3.08 (1.40–6.80), *p* = 0.015, and 4.64 (1.71–12.55), *p* = 0.0002, respectively. Conclusion: Our study confirms the importance of assessing nutritional status in elderly COPD patients and its prognostic value.

## 1. Introduction

Chronic obstructive pulmonary disease (COPD) is a major global health problem characterized by persistent airflow limitation and chronic systemic inflammation, with a significant impact on patients’ overall health and well-being [1]. 

Over the past few decades, there has been sustained research activity in the area of malnutrition, poor quality of life (QoL) and impaired autonomy in activities of daily living (ADLs) in COPD patients. This new area of interest adds relevant information to the comprehensive assessment and management of elderly multimorbid COPD patients, who are poorly represented in clinical trials.

Malnutrition, often suspected because of a low body mass index, is a common problem in COPD patients, with its prevalence ranging from 17% to 47.2% and a pooled prevalence of 30.0%, and the pooled prevalence of being at risk of malnutrition in patients with COPD is 50.0% [2,3,4]. The term malnutrition refers to a condition characterized by an imbalance in energy, protein or other nutrients that leads to adverse effects on body composition, physical function and clinical outcomes [5]. In patients with one or more chronic diseases, such as COPD, malnutrition is a major risk factor for the development of sarcopenia [6]. Poor nutritional status is a major negative determinant of muscle energetics, exercise tolerance and ultimately worsening respiratory symptoms in COPD [7]. There is no validated gold-standard diagnostic tool for evaluating malnutrition in COPD; the Mini Nutritional Assessment (MNA) is a widely used questionnaire able to provide physicians with several types of information about the nutritional status of elderly people. Recent studies show a significant correlation between the nutritional status of COPD patients as evaluated by the MNA and the progression and prognosis of the disease [8], as well as between the former and subjective perception of dyspnea [9]. Evidence suggests that malnutrition, as detected by the MNA score, is associated with several health-related outcomes, including morbidity and mortality [10] but despite the potential of MNA in identifying malnutrition and its associated adverse outcomes in COPD patients, no studies to date have explored the impact of nutritional interventions guided by the MNA on COPD outcomes.

Malnutrition, especially in the elderly, has an intricate relationship with impaired autonomy in ADLs, ultimately leading to a poor QoL; COPD subjects, often in the more advanced stages of disease, show increasing difficulty in performing ADLs, as evidenced by studies that have found impaired self-reported ADL task performance [11,12,13]. This progressive impairment may be due to several factors: the limitation of physical abilities brought on by breathlessness, fatigue and reduced exercise tolerance, which is a common feature of COPD [14,15]; reduced lean mass, respiratory muscular mass, limited ventilatory response and dynamic hyperinflation [7,16]; or the psychological impact of physical decline [17]. Malnutrition-related worsening of lung function and respiratory symptoms may have a negative impact on quality of life (QoL) [18], with implications for exacerbation frequency, hospital admissions and mortality [19,20].

All these factors may have potentially relevant implications for COPD outcomes, especially for the rate and severity of exacerbations, which are periods where symptoms become worse and patients have a greater level of dependence on others for ADLs, experiencing an acute increase in functional impairment and requiring assistance for even basic needs [21,22]. Despite this, there are limited data on the role of poor nutritional status and its secondary effects on limitations in ADLs or the effects of reduced autonomy on clinical outcomes in COPD. So, the aim of the present study was to evaluate the impact of malnutrition, as assessed by the administration of the Mini Nutritional Assessment (MNA), in a cohort of highly complex elderly COPD patients, as a potential prognostic indicator of COPD outcomes.

## 2. Materials and Methods

We consecutively recruited 120 patients with COPD who were referred to the “Internal Medicine and Stroke Care” Unit and the “Cardiovascular Risk” Unit of the Department of Promoting Health, Maternal-Infant, Excellence and Internal and Medicine (Promise) of the Policlinico Paolo Giaccone of the University of Palermo from 1 September 2021 to 1 January 2024. The COPD patients evaluated in this study are part of the larger cohort of the MACH (Multidimensional Approach for COPD and High Complexity) study (NCT04986332). The objectives, materials and methods and main outcomes of the MACH study have been described elsewhere [23]. The MACH study was formally approved by our ethics committee (Comitato Etico Palermo 1; Approval Ref. N. 04/2021, date of approval: 28 April 2021).

Each participant considered in the present analysis underwent a 12-month follow-up, as follows: Participants admitted during hospitalization were reassessed at 3, 6 and 12 months after discharge at the “COPD and Cardiovascular Risk” outpatient clinic, which collected information on both moderate and severe acute COPD exacerbations leading to hospitalization and mortality.For outpatients referred to the “COPD and Cardiovascular Risk” ambulatory, information on moderate and severe acute COPD exacerbation that led to hospitalization was retrospectively collected the day after the last moderate or severe AECOPD and follow-up continued until a 12-month follow-up was completed.

### 2.1. COPD Assessment and Outcome Definition

COPD was diagnosed according to the latest current GOLD (Global Initiative for Chronic Obstructive Lung Disease) report “Global Strategy for Prevention, Diagnosis and Management of COPD” [1]. For participants with a history of COPD, only spirometry tests within six months of enrollment were collected. For participants meeting all the inclusion criteria who had never had a pulmonary function test, outpatient spirometry was performed once the participants were considered clinically stable and free of respiratory infection. Spirometry was performed using the PONY FX tabletop spirometer (COSMED Srl, Rome, Italy). Forced vital capacity (FVC) maneuvers were performed according to the American Thoracic Society/European Respiratory Society Standardisation of Spirometry 2019 Update [24], and the Global Lung Initiative (GLI) reference equation for spirometry [25] was used for comparison with the healthy population. The Modified British Medical Research Council (mMRC) scale and COPD Assessment Test (CAT™) were used to assess symptoms under stable COPD conditions. The primary outcome of the study was a composite of moderate or severe COPD exacerbations during 12 months of follow-up. According to the latest international recommendations [1], a moderate exacerbation was defined as a worsening of respiratory symptoms requiring treatment with a short-acting bronchodilator and oral corticosteroids or antibiotics. A severe exacerbation was defined as hospitalization or an emergency department visit due to worsening respiratory symptoms. Death was also recorded during the study: if an acute exacerbation ended in death, the event was counted as our primary outcome; if death occurred unrelated to acute exacerbations, the event was treated as right-censored. 

### 2.2. Administration of the Questionnaires

The following questionnaires were administered by a professionally trained research assistant:Mini Nutritional Assessment (MNA): The MNA includes anthropometric measurements, a global assessment, a dietary questionnaire and a subjective assessment. According to the developers’ instructions, the MNA’s administration utilizes a two-step approach: the screening step and the global assessment step. Subsequently, based on the MNA Total Score (MNA-TS), patients are classified as ‘‘malnourished’’, ‘‘at risk of malnutrition’’ or as having a ‘‘normal nutritional status”. The ‘‘global assessment step’’ of the MNA should only be administered to patients not reaching the screening threshold. For the purpose of this study, we evaluated both procedures for the MNA questionnaire.The Barthel Index: The evaluation of activities of daily living is essential in gaining insight into the functional capacity and independence of COPD patients. The Barthel Index plays a crucial role in assessing the functional status of these individuals [26]. The Barthel Index, formerly the Maryland Disability Index, was codified by the English nurse Barthel in the 1950s. It serves as an ordinal scale and consists of 10 items examining ADLs. Each item is given an arbitrary score of 5, 10 or 15 points. The sum indicates the degree of autonomy in performing daily activities: a total of 100 points represents the maximum level of autonomy [27]. For the purpose of this study, the cut-off points suggested by Shah et al. [28] were used and allowed us to interpet the Barthel Index score as follows: a total score ranging between 0 and 20 implies “total dependency”, 21–60 indicates “severe dependency”, 61–90 indicates “moderate dependency” and 91–99 suggests “slight dependency”. A score of 100 denotes complete independence from external assistance.EuroQol 5D 3 level (EQ-5D-3L): EQ-5D-3L is a simple questionnaire that explores QoL and health status [29]. Euro-QoL-5D-3L (EQ-5D-3L) is a widely used generic health-related QoL (HRQoL) instrument that measures individuals’ health status across five dimensions: mobility, self-care, usual activities, pain/discomfort and anxiety/depression. It provides a descriptive profile of health and allows for the calculation of an overall index score [29]. Studies using EQ-5D-3L in COPD patients have consistently found that they experience significant impairments in several dimensions of health: key findings in COPD patients include decreased mobility due to breathlessness, limitations in self-care activities and difficulties in performing usual activities. Furthermore, COPD patients often report moderate to severe levels of pain/discomfort and anxiety/depression, which can significantly impact their overall HRQoL [30,31]. For the purpose of this study, the Italian population-based set value was used to calculate the EQ-5D-3L index value [32] (license agreement number: 159432; March 2021).

### 2.3. Statistical Analysis

All the data collected were subjected to statistical analysis of quantitative and qualitative data, including descriptive statistics. Variables were evaluated using histogram and analytical methods (the Shapiro–Wilk test) to determine their distribution. Continuous data are defined as the mean ± standard deviation or median (interquartile range (IR)) if normally distributed or non-normally distributed, respectively. Categorical variables are expressed as frequency counts and percentages. The Kruskal–Wallis test was performed to assess any significant differences in the distribution of continous variables between the three different groups based on the MNA, both Short Form and Total Score. Subsequently, pairwise comparisons were performed using Dunn’s (1964) procedure. Bonferroni correction for multiple comparisons was made with statistical significance accepted at the *p* < 0.0016 level. A multivariate linear regression model was utilized to assess the relationship between the dependent variables (mMRC scale, CAT, the Barthel Index, EQ-5D-3L index value) and independent variables (MNA-SF and MNA-TS). The Fine and Gray subdistribution hazard model was used to estimate the association between the MNA malnutrition risk class and the primary outcome. The competing risk model was chosen to ensure that death unrelated to an acute exacerbation, which could preclude the event of interest, was appropriately accounted for in the regression model. By censoring these cases, we aimed to obtain a more accurate estimate of the cumulative incidence function for COPD exacerbations. All the regression models were adjusted for the following confounding variables: age, sex (female as the reference group), CAT score (<10 points as the reference group) and GOLD class (class I as the reference group). 

A two-tailed *p* value < 0.05 was considered significant. A 95% confidence interval (CI) was reported. The statistical analysis was performed with STATA statistical software, version 18.5 (Stata-Corp, College Station, TX, USA).

## 3. Results

All 120 COPD patients included in this analysis completed the study protocol. Sixteen participants (13.3% of the total sample) died during the follow-up: ten from a lower respiratory tract infection complicated by respiratory failure and the remaining six from complications unrelated to acute exacerbations of COPD.

### 3.1. Characteristics of Demographic, Anthropometric and COPD-Related Variables of the 120 Enrolled Participants Assessed at Baseline

The mean age (SD) was 72.08 years (6.07), and 62.5% were males. The majority of the participants were overweight (34.17%), and their baseline functional impairment due to dyspnea measured through the mMRC scale was 2 to 3 out of 4. Finally, our cohort were characterized by moderate to severe COPD (GOLD class II, 51.67%). All the COPD participants belonged to the “E” group according to the inclusion criteria of “≥2 moderate exacerbations or ≥1 leading to hospitalization”, as recently suggested by the 2024 GOLD report [1]. All the baseline characteristics of the study participants are presented in Table 1.

### 3.2. Distribution of Multidimensional Tests Evaluated: The MNA, the Barthel Index and the EQ-5D-3L Index 

The median value (IR) of the MNA-SF score was 11 (8–12), and according to the “screening score”, 39 participants (32.50%) had a normal nutritional status, 57 (47.5%) were at risk of malnutrition and 24 (20%) were malnourished. The median value (IR) of the MNA-TS score was 22.5 (18.25–24.5), and according to the “Malnutrition Indicator Score”, 45 participants (37.50%) had a normal nutritional status, 62 (51.67%) were at risk of malnutrition and 13 (10.83%) were malnourished. The median value (IR) of the Barthel Index was 90 (77.5–100), and according to Shah et al.’s cut-off [28], the level of functional disability of our cohort was 44 participants (33.67%) with “none” and 17 (14.17%) identified as “slightly dependent”, 42 (35%) as “moderately dependent” and 17 (14.17%) as “severely dependent”. Finally, the median (IR) of the EQ-5D-3L index score was 0.82 (0.72–0.89) out of 1.

### 3.3. Assessment of the Relationship between MNA Indices, Clinical–Spirometric Parameters of COPD Severity, Multidimensional Assessment (The Barthel Index and EQ-5D-3L), Age and BMI

The results of the correlation analysis are shown in Table 2. Our results clearly show that the MNA indices, both Short Form and Total Score, are correlated with the severity of breathlessness, as assessed by the mMRC scale and CAT, and with spirometric variables, specifically with the severity of airflow limitation based on the value of FEV_1_. Also, they were correlated with QoL as measured by EQ-5D-3L and with the degree of independence (only MNA-TS). 

### 3.4. Distribution of mMRC, CAT, FVC, FEV_1_ and EQ-5D-3L Scores According to MNA Risk Class of Malnutrition: “Normal Nutritional Status”, “Risk of Malnutrition” or “Malnourished”

The results are shown in Table 3. The mMRC, CAT, FVC, FEV_1_ and EQ-5D-3L scores were statistically significantly different between the different nutritional statuses based on the MNA-SF: “normal nutritional status” (n:39), “risk of malnutrition” (n:57) and “malnourished” (n:24). The post hoc analysis revealed statistically significant differences in (1) the mMRC scale score distribution of “normal nutritional status” and “at risk of malnutrition” vs. “malnourished”; the same findings were documented for the CAT and EQ-5D-3L distributions; (2) for FVC and FEV_1_, statistically significant differences were found between “normal nutritional status” and “malnourished”; the mMRC, CAT, FVC/FEV_1_, BI and EQ-5D-3L scores were also statistically significantly different between the different nutritional statuses based on the MNA-TS: “normal nutritional status” (n:45), “risk of malnutrition” (n:62) and “malnourished” (n:13). The post hoc analysis revealed statistically significant differences (1) in the mMRC and CAT score distribution of “normal nutritional status” and “at risk of malnutrition” vs. “malnourished”; (2) for FVC/FEV_1_ and BI, between “normal nutritional status” and “malnourished”; and (3) between all groups for the EQ-5D-3L distribution. The Kruskal–Wallis test was used to analyze the distribution of the items examined, except for the comparison of the number of events, where the Chi-square test was used.

### 3.5. Both MNA-SF and MNA-TS Have Predictive Value for the Severity of Breathlessness, Quality of Life and Level of Independence

A multivariate regression analysis was performed to predict the mMRC scale score (model 1), the CAT score (model 2), the BI (model 3) and the EQ-5D-3L index value (model 4) from the MNA indices, adjusting the data for confounding variables. Regression coefficients (β) with 95% confidence intervals, R^2^ and adjusted R^2^ (aR^2^) are shown. Both the MNA-SF and MNA-TS were predictors of breathlessness severity as assessed by the mMRC score and CAT, with QoL measured with EQ-5D-3L and with the degree of independence assessed through the BI. The results are shown in Table 4. The confounding variables considered for each model are fully displayed in Appendix A. 

### 3.6. COPD Participants Classified as “At Risk of Malnutrition” and “Malnourished” According to the MNA Score Had a Higher Risk of Moderate to Severe Acute Exacerbations during the 52 Weeks of Follow-Up

A total of 45 participants experienced exacerbations during the follow-up, and 5 competing events occurred. Competing risk regression according to nutritional status based on the MNA score showed that COPD participants “at risk of malnutrition” and “malnourished” were at a higher risk of moderate to severe acute exacerbations during the 52 weeks of follow-up (respectively, estimated sub-hazard ratios (95% CI): 3.77 (1.29–11.00), *p* = 0.015; 6.12 (2.63–21.21), *p* = 0.002). The same results were obtained performing the analysis according to nutritional status based on the MNA Total Score (Table 5), with a sub-hazard ratio of 3.08, 95% CI 1.40–6.80, *p* = 0.005, for “at risk of malnutrition” and a sub-hazard ratio of 4.64, 95% CI 1.71–12.55, *p* = 0.002, for “malnourished”. The results are shown in Table 5. The confounding variables considered for each competing risk model are fully displayed in Appendix A.

## 4. Discussion

Our study, derived from analysis of the data collected during the follow-up of the ongoing MACH (Multidimensional Approach for COPD and High Complexity) study, demonstrates for the first time that in elderly subjects with COPD and a high burden of comorbidities, assessment of nutritional status by systematic administration of the Mini Nutritional Assessment (MNA) questionnaire allows for the identification of a subgroup of subjects whose poorer nutritional status is significantly associated with several factors of relevant prognostic significance: level of breathlessness severity and respiratory symptoms assessed by the mMRC scale and CAT; severity of airflow limitation based on the values of FEV1 and FVC/FEV1; poorer QoL assessed through the EQ-5D-3 questionnaire; and a lower degree of independence assessed by the Barthel Index. In our data, all the items considered together with nutritional status, severity of airflow limitation (FEV1, FVC/FEV1), clinical severity (mMRC scale and CAT scores), quality of life (EQ-5D-3L index score) and level of dependence (Barthel Index), are also significantly correlated with each other, indicating a high level of frailty, which may require special attention. The possible prognostic relevance is highlighted in the Cox regression analysis suggesting that COPD participants “at risk of malnutrition” and “malnourished” according to nutritional status based on the MNA score have a higher risk of moderate to severe acute exacerbations, with hazard ratios (HRs) of 3.54 (*p* = 0.002) and 8.20 (*p* = 0.0001), respectively, across one year of follow-up.

The natural history of COPD, as well as the rate of global decline in health status, is extremely variable, and the full range of determinants of clinical stability of COPD or its rapid worsening is not fully understood [1,9,20,23]. The identification of additional elements, not directly related to lung disease, that can become targets of intervention to improve clinical stability and reduce the progressive deterioration of health status following recurrent exacerbations is an urgent need in COPD. Malnutrition could be one of the main targets to point out; it is a common finding in patients with COPD [2,3,4,33]. Its prevalence is very variable, ranging from 17% to 47.2%, due to the different types of patients in which it is assessed and the different diagnostic criteria used, but it is clear that the more severe the disease, the higher the risk of malnutrition and sarcopenia. In our study, 47.5% of subjects were classified as at risk of malnutrition, and 20 were malnourished, data consistent with the available evidence and confirming the relevance of this issue. Of note, underweight is not an issue in our sample: the mean BMI is 27.8, and only 3.3% of the total population have a BMI < 18.5 kg/m^2^. Neither BMI nor age are significantly different in different malnutrition classes according to the MNA (Table 2 and Table 3), demonstrating the need for additional routine nutritional screening to detect poor nutritional status in COPD given it is often difficult to suspect malnutrition in this category of subjects based on BMI alone. Another important issue is that the prevalence of malnutrition is reported to be influenced by geographical regions, which may reflect the negative impact of social and economic factors in developing countries [4], but also removing this additional determinant, even in a high-income country such as Italy, the relevance of this aspect and its potential consequences for the natural history of the disease cannot be underestimated.

The clinical implications of poor nutritional status in people with COPD are strictly related to the pathophysiology of the disease [7,34], which often leads to a disease-related energy imbalance, with a hypercatabolic state also inducing low-grade systemic inflammation, tissue hypoxia and ultimately muscle wasting and atrophy. All these events can be further amplified by aging, comorbidities and politherapy. Our data suggest a linear relationship, with progressive worsening between the three MNA risk classes of malnutrition according to the Kruskal–Wallis test (Table 3): the poorer the nutritional status, the worse the lung function and the greater the clinical severity. Similar results were provided by Fekete et al. on 50 subjects in Hungary but using the Malnutrition Universal Screening Tool (MUST) and bioelectrical impedance analysis (BIA) as their screening tools [18]. 

Malnutrition in in our COPD population appears to have a significant impact on the risk of acute exacerbations, quantified as a 3.0 times higher risk of moderate to severe acute exacerbations for patients “at risk of malnutrition” and a 4.6 times higher risk for patients who are “malnourished” along one year of follow-up (Table 5). Only a few studies have attempted to determine the prognostic role of malnutrition, two in stable COPD [19,35] and two during acute exacerbations [36,37], with wide heterogeneity of the results but increased mortality indicated in malnourished COPD patients. These papers adopted the European Society of Clinical Nutrition and Metabolism (ESPEN) criteria for identifying clinical malnutrition [38] or the Global Leadership Initiative on Malnutrition (GLIM) criteria [39], giving a relevant role to BMI, which in our sample appears to be an inaccurate parameter for COPD subjects. No studies to date have provided information on the risk of acute exacerbations during 12 months of follow-up according to screening-based use of the MNA in COPD.

This is not an intervention study, but our study suggests that targeted strategies to improve nutritional status in elderly COPD may have relevant prognostic value. As recommended by the latest GOLD report [1], a combination of exercise, specific interventions to control respiratory and systemic inflammation and targeted nutritional support may be used to prevent all the negative effects of the development of pulmonary cachexia. Future developments may consider the use of nutritional supplements for this category of subjects, such as oral supplementation with ketone bodies, as already proposed for subjects with heart failure [40]. Over dietary intervention, exercise, especially in comorbid patients where cardiovascular, metabolic and respiratory disorders coexist, is recognized as the better intervention strategy to reverse some of the skeletal muscle abnormalities typical of COPD patients [41,42], being the most effective non-pharmacological intervention to improve exercise capacity and dyspnea. Systemic inflammation and oxidative stress, which have been postulated to be etiological factors of muscle dysfunction in COPD, may be also a therapeutic target through an integrated approach combining nutrition, exercise and drugs. The MACH (Multidimensional Approach for COPD and High Complexity) study, from which these represent preliminary data, will also evaluate these aspects in an ongoing intervention arm of the study.

Our study has some limitations, with the first being the reliability of the MNA as a screening tool in COPD; data show that the measurement tool influenced the malnutrition prevalence and the “at risk of malnutrition” prevalence among patients with COPD [4]. To date, there is still a lack of a gold-standard diagnostic tool for evaluating malnutrition in COPD. Since malnutrition is a very complex subject, the attempt to validate a world-wide gold standard for malnutrition is challenging, and this assumption is even more valid for COPD. The MNA is one of the most widely used tests worldwide, and the present analysis in our opinion could be useful to validate its use in elderly high-complexity COPD subjects. Second, our sample consisted of very complex COPD patients with a high degree of comorbidities [23], so our results may not be generalizable to the entire clinical spectrum of COPD. Given this, this category of patients with the highest level of frailty is the one that may benefit more from a multidimensional assessment, which needs to include a validated tool to screen nutritional status. Thirdly, the use of a screening tool such as the MNA is subject to the judgment of the operator and requires the active co-operation of the patients, as the information provided may be subjectively transmitted and subjectively received, which reduces the reproducibility of the data.

## 5. Conclusions

In elderly COPD patients at high risk of exacerbations, systematic screening for malnutrition using the MNA identifies categories of patients at different risks of acute exacerbations: the poorer their nutritional status, the higher the risk. Inadequate nutritional health is significantly associated with a worse clinical profile, lung function and perceived quality of life. BMI per se may not be an accurate parameter of good/bad nutritional status in COPD. 

Our study confirms the importance of a multidimensional assessment in elderly patients with a high burden of comorbidities, as several determinants have a strong influence on the clinical course of respiratory disease. In particular, the MNA may also provide prognostic value; subjects at risk of malnutrition or with overt malnutrition in our analysis have a higher risk of moderate to severe acute exacerbations.

Poor nutritional status should merit being subjected to a targeted clinical pathway that addresses all the clinical and social vulnerabilities leading to malnutrition with an individualized treatment plan.

## Figures and Tables

**Table 1 nutrients-16-02303-t001:** Demographic, anthropometric and COPD-related variables at baseline.

Variable		Count (%)	Mean
Male, n (%)		72 (62.5)	
Age (years), median (IQR)			73 (67–79)
Former smoker (yes), n (%)		75 (62.50)	
Active smoker (yes), n (%)		45 (37.50)	
Environmental risk factors (yes), n (%)		37 (30.83)	
BMI (kg/m^2^), median (IQR)			27.8 (24.2–31.2)
Obesity class according to BMI	Underweight, n (%)	4 (3.33)	
	Optimal weight, n (%)	32 (26.67)	
	Overweight, n (%)	41 (34.17)	
	Class I obesity, n (%)	30 (25)	
	Class II obesity, n (%)	9 (7.5)	
	Class III obesity, n (%)	4 (3.33)	
mMRC, n (%)	0	5 (4.17)	
	1	25 (20.83)	
	2	32 (26.67)	
	3	39 (32.5)	
	4	19 (15.83)	
CAT, mean ± SD			15.94 ± 7.95
FVC (Lt), mean ± SD			2.33 ± 0.72
FVC (% predicted), median (IQR)			72 (60–86)
FEV_1_ (Lt/sec), median (IQR)			1.4 (1.01–1.85)
FEV_1_ (% predicted), mean ± SD			58.33 ± 18.90
FEV_1_/FVC, median (IQR)			0.63 (0.54–0.68)
COPD-GOLD class, n (%)	GOLD 1E	14 (11.67)	
	GOLD 2E	62 (51.67)	
	GOLD 3E	38 (31.67)	
	GOLD 4E	6 (5)	
Inhaled bronchodilators, n (%)	LAMA	22 (18.33)	
	LABA + ICS	20 (16.67)	
	LABA + LAMA	37 (30.83)	
	LABA + LAMA + ICS	41 (34.17)	
LTOT, n (%)		39 (32.50)	
Outpatient enrollment, n (%)		70 (58.33)	

Data are presented as mean values ± one standard deviation (SD) or as medians and interquartile ranges (IQRs); BMI: body mass index; mMRC: modified Medical Research Council; CAT: COPD Assessment Test; FVC: forced vital capacity; FEV_1_: forced expiratory volume in the first second; COPD: chronic obstructive pulmonary disease; GOLD: Global Initiative for Chronic Obstructive Lung Disease; LAMA: long-acting muscarinic antagonist; LABA: long-acting β-agonist; ICS: inhaled corticosteroid; LTOT: long-term oxygen therapy; obesity classes are defined as “Underweight” for BMI < 18.5 kg/m^2^, “Optimal weight” for BMI ranging from 18.5 to 24.9 kg/m^2^, “Overweight” for BMI ranging from 25 to 29.9 kg/m^2^,“Class I Obesity” for BMI ranging from 30.0 to 34.9 kg/m^2^, “Class II Obesity” for BMI ranging from 35 to 39.9 kg/m^2^, “Class III Obesity” for BMI ≥ 40.0 kg/m^2^.

**Table 2 nutrients-16-02303-t002:** Relationship between MNA indices and the main clinical–spirometric parameters of COPD severity.

		mMRC	CAT	FVC (%)	FEV_1_ (%)	Barthel Index	EQ-5D-3L	Age	BMI	MNA-SF	MNA-TS
MNA-SF	ρ	−0.380 ***	−0.414 ***	0.247 *	−0.312 ***	0.147	0.424 ***	−0.07	0.140	--	0.831 ***
MAN-TS	ρ	0.398 ***	0.448 ***	0.126	−0.267 **	0.224 **	0.494 ***	−0.02	0.132	0.831 ***	--

mMRC: modified Medical Research Council; CAT: COPD Assessment Test; FVC: forced vital capacity; FEV_1_: forced expiratory volume in the first second; EQ-5D-3L: EuroQol 5D three-level; BMI: body mass index; MNA-SF: Mini Nutritional Assessment—Short Form; MNA-TS: Mini Nutritional Assessment—Total Score; *** *p* < 0.001; ** *p* < 0.01; * *p* < 0.05.

**Table 3 nutrients-16-02303-t003:** Distribution of main clinical–spirometric variables and questionnaires administered according to MNA risk class of malnutrition.

	Mini Nutritional Assessment Short Form				Mini Nutritional Assessment Total Score			
	Normal (1) (n = 39)	At Risk (2) (n = 57)	Malnourished (3) (n = 24)	*p* Value	Normal (1) (n = 45)	At Risk (2) (n = 62)	Malnourished (3) (n = 13)	*p* Value
Age	71 (66–78)	74 (68–80)	73.5 (66–79.5)	NS	74 (67–78)	73 (67–80)	71 (70–78)	NS
BMI	27.5 (24.2–30.9)	28.7 (24.5–31.2)	25.7 (22–33.5)	NS	28.3 (24.6–31.1)	27.5 (24.1–31.2)	25.1 (21.6–34.15)	NS
mMRC	2 (1–3)	3 (2–3)	3 (2–4)	1 vs. 2–3, *p* < 0.01	2 (1–3)	3 (2–3)	3 (3–4)	1 vs. 2–3, *p* < 0.01
CAT score	11 (5–15)	18 (13–22)		1 vs. 2–3, *p* < 0.001	13 (6–17)	17 (11–22)	23 (16–28)	1 vs. 2–3, *p* < 0.001
FVC (%)	77.5 (65–91.5)	75 (60–86)	62 (56–73)	1 vs. 3, *p* = 0.019	75 (61–88.5)	71.5 (58–87)	71 (59–78)	NS
FEV_1_ (%)	66 (51–78)	59 (40–68)	50 (39–59)	1 vs. 3, *p* = 0.01	62 (49–74)	58 (44–68)	44 (34–55)	NS
FVC/FEV_1_	0.65 (0.59–0.69)	0.62 (0.51–0.68)	0.62 (0.58–0.65)	NS	0.65 (0.59–0.69)	0.63 (0.54–0.68)	0.52 (0.45–0.62)	1 vs. 3, *p* = 0.002
Barthel Index	95 (85–100)	90 (75–100)	92.5 (67.5–100)	NS	95 (85–100)	92.5 (80–100)	75 (45–90)	1 vs. 3, *p* = 0.015
EQ-5D-3L	0.88 (0.81–0.92)	0.78 (0.7–0.88)	0.72 (0.52–0.82)	1 vs. 2–3, *p* < 0.01	0.88 (0.81–0.9)	0.78 (0.72–0.87)	0.56 (0.37–0.72)	All comparisons *p* < 0.001
MNA-SF	12 (12–14)	10 (9–11)	7 (5.5–7)	All comparisons *p* < 0.0001	12 (11–14)	10 (8–11)	6 (5–7)	All comparisons *p* < 0.001
MNA-TS	25 (24–27)	22 (19.5–23.5)	17.25 (15.25–18.5)	All comparisons *p* < 0.0001	25 (24–27)	20 (18–22.5)	15 (14–16)	All comparisons *p* < 0.0001
Time to event (months)	52 (52–52)	52 (19–52)	27.5 (12–47)	All comparisons *p* < 0.05	52 (52–52)	47.5 (11–52)	15 (11–35)	All comparisons *p* < 0.05
Number of events, n (%)	5 (11.11)	25 (55.56)	15 (33.33)	1 vs. 2–3, *p* < 0.001 ^†^	8 (17.78)	29 (64.44)	8 (17.78)	1 vs. 2–3, *p* < 0.001 ^†^

Data are presented as medians (interquartile ranges); BMI: body mass index; mMRC: modified Medical Research Council; CAT: COPD Assessment Test; FVC: forced vital capacity; FEV_1_: forced expiratory volume in the first second; MNA-SF: Mini Nutritional Assessment—Short Form; MNA-TS: Mini Nutritional Assessment—Total Score; EQ-5D-3L: EuroQol 5D three-level; NS: not significant; ^†^ Test based on Chi-square test.

**Table 4 nutrients-16-02303-t004:** Results of the multivariate regression analysis for mMRC (model 1), CAT (model 2), BI (model 3) and EQ-5D-3L (model 4).

	MNA-SF β Coefficient	MNA-TS β Coefficient	95% Confidence Interval
Model 1.1	−0.135 ***		−0.207–−0.064
Model 1.2		−0.095 ***	−0.140–−0.050
Model 2.1	1.075 ***		−1.607–−0.549
Model 2.2		−0.844 ***	−1.172–−0.517
Model 3.1	1.424 *		0.062–2.788
Model 3.2		1.605 ***	0.766–2.445
Model 4.1	0.273 ***		0.146–0.400
Model 4.2		0.231 ***	0.015–0.030

* *p* < 0.05; *** *p* < 0.001; an increase of 1 point is intended for continuous variables, except for model 4 (0.1-point increase); MNA-SF: Mini Nutritional Assessment—Short Form; MNA-TS: Mini Nutritional Assessment—Total Score.

**Table 5 nutrients-16-02303-t005:** Risk of moderate to severe acute exacerbations according to nutritional status (MNA score).

	Normal Nutritional Status	At Risk of Malnutrition	Malnourished
MNA Short Form	Group reference	3.77 (1.29–11.00) *	6.12 (2.63–21.21) *
MNA Total Score	Group reference	3.08 (1.40–6.80) **	4.64 (1.71–12.55) **

Results are presented as sub-hazard ratios with 95% confidence intervals (CIs) and *p* values. ** *p* < 0.01; * *p* < 0.05.

## Data Availability

The full dataset is available upon request from the corresponding author: the data are part of the ongoing MACH study, which is still in follow-up.

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
