# Peer review of "The Impact of Malnutrition on Chronic Obstructive Pulmonary Disease (COPD) Outcomes: The Predictive Value of the Mini Nutritional Assessment (MNA) Versus Acute Exacerbations in Patients with Highly Complex COPD and Its Clinical and Prognostic Implications"

_nutrients, 2024, doi:10.3390/nu16142303_

Round 1
Reviewer 1 Report
Comments and Suggestions for Authors
The study provides an analysis of nutritional status in COPD outcomes based on acute exacerbations. While the results are interesting, the presentation of results and statistical analyses needs improvement. The main areas of concern are;
1. No data is given for the primary outcome of COPD exacerbations during the study. The number of exacerbations per participant, time of first exacerbation should be presented in Table 3 so the reader has an understanding of the data.
2. Sub title os the manuscript should detail what is being presented, not the statistic being used eg 2.3 "results of the Kruskal-Wallis test" should instead described the key finding based on MNA status.
3. The study and statistics don't allow predictions, the authors should becareful with their terminology as associations are instead found.
4. The abstract "multivariate linear logistic models" should be multivariable linear and logistic models". The statistics are univariable and multivariable and not univariate /multivariate.
5. Confidence intervals should be given the in the abstract and not just the HR.
6. Section 2.3 "indipendent" is spelt incorrectly.
7. As described in the statistical analysis section and in the tables, it is not standard practice to report the R^2 value. This is the not the method to assess the goodness of fit of the models. Other tests should be done to assess the model, but these tests do not need to be reported.
8. It is mentioned that 16 participants died, how were they handled in the analysis?
9. Table 1 should be clarified with the continuous variables, the column title in mean (SD) while the variable has median (IQR). it should be made clear, and the column title should be changed to include both.
10. Table 1 "theraphy" is spelt incorrectly
11. Supplementary Table 4, should be "4"
12. Table 4, R62 values should be removed and it is more important to have the CI for B coefficients.
13. I'm not sure why the Cox regression is done as both a categorical variable of MNA groups and as a continuous variable. Given the MNS identifies risk categories, I don't see the value doing the analysis a a continuous variable gives.
Author Response
The study provides an analysis of nutritional status in COPD outcomes based on acute exacerbations. While the results are interesting, the presentation of results and statistical analyses needs improvement. The main areas of concern are;
- No data is given for the primary outcome of COPD exacerbations during the study. The number of exacerbations per participant, time of first exacerbation should be presented in Table 3 so the reader has an understanding of the data.
Thank you for your comment and your kind suggestion. In the revised version, we have updated the information presented in Table 3 by adding the number of exacerbations recorded during follow-up and the mean time to first exacerbation. We have also added information on the number of events observed in paragraph 3.6.
- Subtitle of the manuscript should detail what is being presented, not the statistic being used eg 2.3 "results of the Kruskal-Wallis test" should instead described the key finding based on MNA status.
Thank you for your helpful and appropriate suggestion. In the revised version, we are changing the subtitle headings to describe the objective of the analysis being described, rather than the analysis itself.
- The study and statistics don't allow predictions, the authors should be careful with their terminology as associations are instead found.
We fully agree that terminology is crucial for the correct presentation of results to avoid misunderstandings for the reader. Accordingly, we are revising the paper and deleting any reference to a possible "predictive role" of MNA or other items considered in the present analysis. We have also slightly changed the title of the paper; the revised version refers to the "predictive value" of MNA, which has a more precise statistical value and refers to "the likelihood of accurately determining the occurrence or non-occurrence of an event".
- The abstract "multivariate linear logistic models" should be multivariable linear and logistic models". The statistics are univariable and multivariable and not univariate /multivariate.
Thank you for reporting this spelling error. We performed a multiple linear regression analysis. the abstract has been amended accordingly.
- Confidence intervals should be given the in the abstract and not just the HR.
Thank you for the suggestion; in the abstract the sub-hazard ratios are now presented with confidence intervals.
- Section 2.3 "indipendent" is spelt incorrectly
Thank you for pointing out this spelling error, which has now been corrected in the revised version of the paper.
- As described in the statistical analysis section and in the tables, it is not standard practice to report the R^2 value. This is the not the method to assess the goodness of fit of the models. Other tests should be done to assess the model, but these tests do not need to be reported.
We agree with the reviewer on this point. The mention of the coefficient of determination (R2) has been removed in the revised version of the statistical analysis section.
- It is mentioned that 16 participants died, how were they handled in the analysis?
Thank you for the important comment. We have added a description in the Methods section (paragraph 2.1) of how the fatalities that occurred during the follow-up were handled.
- Table 1 should be clarified with the continuous variables, the column title in mean (SD) while the variable has median (IQR). it should be made clear, and the column title should be changed to include both.
Thank you for your comments on this issue. We have changed Table 1 to alternatively report whether mean + standard deviation or median and interquartile range was displayed.
- Table 1 "theraphy" is spelt incorrectly
Thank you for pointing out this spelling error, which has now been corrected in the revised version of the paper.
- Supplementary Table 4, should be "4"
We have corrected the table numbering in the supplementary material as Supplementary Table 1 and so on
- Table 4, R62 values should be removed and it is more important to have the CI for B coefficients.
Thank you for your comment. In the first draft, we prefer to keep the R2 in Table 4 to give the reader an idea of the proportion of variability in the dependent variable that is explained by the regression model; following the reviewer's suggestion, we have removed this information in the revised version. R2 values can be found in the full models presented in Supplementary Table 1.
- I'm not sure why the Cox regression is done as both a categorical variable of MNA groups and as a continuous variable. Given the MNS identifies risk categories, I don't see the value doing the analysis a a continuous variable gives.
Thank you for the accurate remark. As pointed out, it is not appropriate to perform an analysis treating as a continuous variable a test originally derived to identify risk classes. We have decided to delete the previous analysis based on the Cox regression, which was methodologically incorrect, and to perform a competing-risks regression analysis according to the nutritional status which is now shown in table 5. All the paper was corrected according to the new analysis.
Reviewer 2 Report
Comments and Suggestions for Authors
This is an interesting manuscript that discusses the association between nutrition management and COPD status. The authors applied various statistical analyses to various clinical measurements. However, this manuscript is lacking with a clear description of results explanation. If the author can provide a better explanation of how the results describe the hypothesis or at least not using the statistical analysis method name as the subtitle of subsection.
Also, please check the format of the table, subtitle numbers, and table legends; it is not matching across the manuscript.
Author Response
This is an interesting manuscript that discusses the association between nutrition management and COPD status. The authors applied various statistical analyses to various clinical measurements. However, this manuscript is lacking with a clear description of results explanation. If the author can provide a better explanation of how the results describe the hypothesis or at least not using the statistical analysis method name as the subtitle of subsection.
Also, please check the format of the table, subtitle numbers, and table legends; it is not matching across the manuscript.
Thank you for highlighting this area for improvement in the paper. We have tried to describe our results in a clearer and more detailed way. This is to reach out to the reader and to better discuss our findings.
We have changed all the titles of subsections to describe the main objective of the analysis performed rather than the analysis itself.
We have checked the entire manuscript for formatting and table legends. Unfortunately, due to the characteristic of the analysis presented, it has not been possible to achieve a consistent format for all the tables; in some cases we have favoured the readability over the graphics. We hope for your understanding.
Round 2
Reviewer 1 Report
Comments and Suggestions for Authors
The authors have addressed my concerns.
Author Response
The authors have addressed my concerns
Thank you for your helpful comments and detailed suggestions, which were essential in improving the quality and presentation of our manuscript.
Reviewer 2 Report
Comments and Suggestions for Authors
Thanks for the authors to address the comments. However, the authors still not direct revise the sub title to describe the major finding driven from the results. For example, subtitle 3.5.. Changing the subtitle to distribution of multivariate analysis is not acceptable as a sub title in results sections rather than as a subtitle for method sections.
Please strongly consider summarizing the finding of the result sections as the subtitle name.
The rest are fine with me.
Author Response
Thanks for the authors to address the comments. However, the authors still not direct revise the sub title to describe the major finding driven from the results. For example, subtitle 3.5.. Changing the subtitle to distribution of multivariate analysis is not acceptable as a sub title in results sections rather than as a subtitle for method sections.
Please strongly consider summarizing the finding of the result sections as the subtitle name.
The rest are fine with me.
Thank you for further highlighting this area for improvement in the paper. Following your suggestion, we have revised the titles of all subsections of the Results section and, in particular, in paragraphs 3.5 and 3.6, we have used the subtitle to summarize the main finding of the result described.